# Design and Immune Profile of Multi-Epitope Synthetic Antigen Vaccine Against SARS-CoV-2: An In Silico and In Vivo Approach

**DOI:** 10.3390/vaccines13020149

**Published:** 2025-01-31

**Authors:** Maria da Conceição Viana Invenção, Larissa Silva de Macêdo, Ingrid Andrêssa de Moura, Lucas Alexandre Barbosa de Oliveira Santos, Benigno Cristofer Flores Espinoza, Samara Sousa de Pinho, Lígia Rosa Sales Leal, Daffany Luana dos Santos, Bianca de França São Marcos, Carolina Elsztein, Georon Ferreira de Sousa, Guilherme Antonio de Souza-Silva, Bárbara Rafaela da Silva Barros, Leonardo Carvalho de Oliveira Cruz, Julliano Matheus de Lima Maux, Jacinto da Costa Silva Neto, Cristiane Moutinho Lagos de Melo, Anna Jéssica Duarte Silva, Marcus Vinicius de Aragão Batista, Antonio Carlos de Freitas

**Affiliations:** 1Laboratory of Molecular Studies and Experimental Therapy—LEMTE, Department of Genetics, Federal University of Pernambuco, Recife 50670-901, Brazil; maria.conceicao@ufpe.br (M.d.C.V.I.); larissa.smacedo@ufpe.br (L.S.d.M.); ingrid.andressa@ufpe.br (I.A.d.M.); benigno.cristofer@ufpe.br (B.C.F.E.); samara.pinho@ufpe.br (S.S.d.P.); ligia.leal@ufpe.br (L.R.S.L.); daffany.luana@ufpe.br (D.L.d.S.); bianca.saomarcos@ufpe.br (B.d.F.S.M.); carolina.elsztein@cetene.gov.br (C.E.); anna.jessica@ufpe.br (A.J.D.S.); 2Laboratory of Molecular Genetics and Biotechnology (GMBio), Department of Biology, Center for Biological and Health Sciences, Federal University of Sergipe, São Cristóvão 49100-000, Brazil; luccalexandre@gmail.com (L.A.B.d.O.S.); mvabatista@hotmail.com (M.V.d.A.B.); 3Laboratory of Immunological and Antitumor Analysis, Keizo Asami Immunopathology Laboratory, Department of Antibiotics, Bioscience Center, Federal University of Pernambuco, Recife 50670-901, Brazil; georon.sousa@ufpe.br (G.F.d.S.); guilherme.souzasilva@icb.usp.br (G.A.d.S.-S.); barbara.sbarros@ufpe.br (B.R.d.S.B.); leonardo.oliveiracruz@ufpe.br (L.C.d.O.C.); cristiane.moutinho@ufpe.br (C.M.L.d.M.); 4Laboratory of Cytological and Molecular Research, Department of Histology and Embriology, Federal University of Pernambuco, Recife 50670-901, Brazil; jullianomaux99@gmail.com (J.M.d.L.M.); jacintocosta@hotmail.com (J.d.C.S.N.)

**Keywords:** vaccinology, immunoinformatics, variants

## Abstract

Background: The rapid advancement of the pandemic caused by SARS-CoV-2 and its variants reinforced the importance of developing easy-to-edit vaccines with fast production, such as multi-epitope DNA vaccines. The present study aimed to construct a synthetic antigen multi-epitope SARS-CoV-2 to produce a DNA vaccine. Methods: A database of previously predicted Spike and Nucleocapsid protein epitopes was created, and these epitopes were analyzed for immunogenicity, conservation, population coverage, and molecular docking. Results: A synthetic antigen with 15 epitopes considered immunogenic, conserved even in the face of variants and that were able to anchor themselves in the appropriate HLA site, together had more than 90% worldwide coverage. A multi-epitope construct was developed with the sequences of these peptides separated from each other by linkers, cloned into the pVAX1 vector. This construct was evaluated in vivo as a DNA vaccine and elicited T CD4+ and T CD8+ cell expansion in the blood and spleen. In hematological analyses, there was an increase in lymphocytes, monocytes, and neutrophils between the two doses. Furthermore, based on histopathological analysis, the vaccines did not cause any damage to the organs analyzed. Conclusions: The present study generated a multi-epitope synthetic vaccine antigen capable of generating antibody-mediated and cellular immune responses.

## 1. Introduction

At the beginning of 2020, several countries worldwide reported a high number of cases of infection and death by Severe Acute Respiratory Syndrome Coronavirus 2 (SARS-CoV-2), promoting the disease called COVID-19. As it is a respiratory virus with high transmissibility and mutagenic potential, the production of vaccines became an essential tool in controlling the pandemic [1,2]. Despite the circulation of several variants, the rapid study, development, and establishment of vaccine techniques have helped break the viral virulence against the immune system and its transmissibility [3]. Several licensed vaccines employ strategies such as attenuated virus, viral vector, subunits, and nucleic acid vaccines [4]. Some of these vaccines are very expensive and need high technology for their development. Thus, especially in underdeveloped countries, easy-to-edit and low-cost vaccine strategies have been demanded.

COVID-19 vaccines must be able to elicit especially a Th1 response (IL-2, IL-6, TNF, and IFN), with an increase in CD8 T-cell action and CD4 T and B memory cells and the promotion of anti-SARS-CoV-2 NAb long-term production [1,5]. Moreover, currently, studies have been showing that some SARS-CoV-2 variants, such as Alpha (B.1.1.7), Beta (B.1.351), Gamma (P1), Delta (B.1.617.2), Epsilon (B.1.427/B.1.429), Eta (B.1.525), Lota (B.1.526), and Kappa (B.1.617.1), partially escape humoral but not T-cell responses [5].

Multi-epitope synthetic antigens are promising alternatives that can be built using structural, non-structural (nsPs), and accessory proteins conserved regarding the new virus variants. However, several studies that developed multi-epitope constructs against COVID-19 did not include analysis to assess conservation for emerging variants or immunological assays [6,7,8,9,10,11]. In this context, studies about constructing synthetic antigens from screening epitopes that include in vitro and in vivo data become attractive. It is possible to select epitopes that contribute to increased immunogenicity concerning vaccines under development through accurate data, requiring low-cost tools once analyses can be performed through free software.

Thus, the present study aimed to build and evaluate a synthetic antigen based on T- and B-cell linear epitopes derived from the structural proteins Spike (S) and Nucleocapsid (N) of SARS-CoV-2 from computational analyses as an immunization strategy against COVID-19 based on a multi-epitope DNA vaccine. This study differs from others in that it includes the construction of little-explored in silico approaches, including the assessment of immunogenicity, conservation between different viral variants, safety and stability; but it also presents preliminary in vitro and in vivo data regarding the immune response induced by the vaccine construction.

## 2. Materials and Methods

### 2.1. Protein Sequence Retrieval

The amino acid sequences of a Spike protein (YP_009724390.1) and a Nucleocapsid protein (YP_009724397.2) were obtained from the SARS-CoV-2 reference sequence of the National Center for Biotechnology Information Database [12].

### 2.2. Epitope Prediction and Analysis

The Immune Epitope Database and Analysis Resource (IEDB) [13] was utilized to predict T-cell epitopes restricted to HLA classes I and II from the Spike and Nucleocapsid protein sequences. The TCD8+ (CTL) epitopes for HLA class I alleles (Table 1) were predicted using the “MHC-I Binding Prediction” tool employing the NetMHCpan 4.1 EL recommended binding predictor-2023.09. The TCD4+ (HTL) epitopes for HLA class II alleles (Table 1) were predicted using the “MHC-II Binding Prediction” tool employing the NetMHCIIpan 4.1 EL recommended binding predictor-2023.09. The threshold used for epitopes to proceed to the next analyses was epitopes that bind to more than one HLA with a percentile rank lower than 100 (the lower the percentile, the greater the binding with the HLA). Therefore, the vaccine antigen ultimately not only has a cellular response but also aims to generate a humoral response; B-cell epitopes predicted by Rahman et al. [14] and Dong et al. [6] were selected.

The immunogenicity of CTL epitopes was analyzed using the “Class I Immunogenicity” tool [15], in which higher scores indicate greater immunogenicity, with epitopes with a score > 0 being selected for the next analysis. The potential of epitopes to be toxic was evaluated by the ToxinPred server [16]. Non-toxic epitopes were evaluated for their conservation against SARS-CoV-2 variants obtained from the GISAID (Alpha—B.1.1.7 (ID: 3373203), Beta—B.1.351 (ID: 3369915), Gamma—P.1 (ID: 3373439), Delta—B.1.617.2 (ID: 3372997), Eta—B. 1525 (ID: 3373127), Iota—B.1.526 (ID: 3372470), Kappa—B.1.617.1 (ID: 3356538), Lambda—C.37 (ID: 3370455), Zeta—P.2 (ID: 3373365) and Omicron—B.1.1.529 (ID: 9588977)) and from Park et al. [17] using the “Epitope Conservancy Analysis” tool [18]. The CTL and HTL epitopes that showed conservation above 80% were analyzed for population coverage using the “Population Coverage” tool [19]. The MHC class I and II epitopes that composed the final vaccine construct were analyzed individually and together for their population coverage worldwide and for the different continents.

### 2.3. Molecular Interaction Analysis Among the TCD8+ and TCD4+ Epitopes with HLAs Alleles

The epitope three-dimensional (3D) structures were modeled using the PEP-FOLD 3.5 server [20]. The crystallographic structures of all HLAs that were found available in the PDB were treated by removing the ligands described in Table 1 and the associated water molecules using AutoDockTools 1.5.6 version [21]. The files of the epitope modes and the HLAs treated were docked using AutoDock Vina, available in the PyRx software 0.8 version [22]. The data generated from molecular docking are based on the energy released from the binding of the epitope to the HLA (it is estimated that the lower the free energy, the greater the binding affinity between the molecules). The structure of each docking complex was visualized in a Biovia Discovery Studio Visualizer [23].

The stability of the docked complexes was simulated using atomistic molecular dynamics simulations, using a NAMD 2.14 server and CHARMM36m force field [24,25]. The preparation and configuration of the systems were performed using a CHARMM GUI Solution Builder [26]. The trajectories were analyzed using VMD 1.9 [27], where the stability and behavior of each complex were evaluated through the analysis of the root mean square deviation (RMSD) and root mean square fluctuation (RMSF) of backbone atoms.

### 2.4. Multi-Epitope Vaccine Antigen Design and Antigenicity, Allergenicity, Cross-Reactivity, Solubility, and Physicochemical Properties

The B- and T-cell epitopes were arranged to form the sequence of a multi-epitope vaccine, named Scov2-Sint1. Spacer sequences called linkers, such as AAY [28], were added among the CTL epitopes; GPGPGs [6,29,30] were added among the HTL epitopes; and KKs [31] were added between the B-cell epitopes of the multi-epitope sequence. These spacer sequences were used to delimit the cleavage points at the beginning and end of the epitope and thus form the peptide with the expected dimensions.

After the assembly of the complete multi-epitope sequence, an AlphaFold server [32] predicted the 3D structure that was subsequently refined using a Galaxy Refine server [33] to adjust so that the residues of the protein were positioned at a higher percentage in more favorable regions. The stereochemical quality of the pre- and post-refinement structures was observed through Ramachandran diagrams obtained using the PDBsum server [34].

The potential antigenicity, allergenicity, cross-reactivity, solubility, physicochemical properties, and immune simulation of Scov2-Sint1 were evaluated using the VaxiJen 2.0 predictor [35], AllerTOP v.2.0 [36], BLASTp [37] using as an organism for comparison the genome of Homo sapiens, Protein-Sol server [38], ProtParam server [39], and C-ImmSim server [40], respectively.

### 2.5. Molecular Docking of Multi-Epitope Vaccine and Toll-like Receptors (TLRs)

The docking between Scov2-Sint1 and the immune receptors TLR2 (PDB ID: 2Z7X), TLR3 (PDB ID: 2A0Z), and TLR4 (PDB ID: 4G8A) was simulated by the ClusPro 2.0 server [41]. The result of the docked complexes were visualized by a Biovia Discovery Studio Visualizer [23].

### 2.6. DNA Vaccine Preparation

The Scov-Sint1 antigen was codon-optimized for expression in mammalian cells and synthesized by GenOne Biotechnologies (Rio de Janeiro, Brazil). Restriction sites for the *BamH*I and *Not*I enzymes were inserted to allow the cloning in the pVAX1 (Invitrogen, Carlsbad, CA, USA) mammalian expression vector. A recombinant vector and an empty pVAX1 were used to transform chemically competent *E. coli* Top10 cells [(F-mcrA D(mrr-hsdRMS-mcrBC) ϕ 80lacZD M15 DlacX74 recA1 araD139D (araleu)7697 galUgalKrpsL (StrR) endA1 nupG)] (Invitrogen). Clone screening was performed by plasmid extraction followed by restriction analyses.

*E. coli* transformed with both vectors were expanded in LB medium (Luria-Bertani) for plasmid DNA extraction with a PureLink™ HiPure Plasmid Maxiprep kit (Invitrogen, Carlsbad, CA, USA), according to the manufacturer’s instructions. The obtained DNA was adjusted to a final concentration of 50 µg in 30 µL.

### 2.7. Transfection and Expression Analysis After Cell Culture

HEK 293T cells were maintained in DMEM (Sigma-Aldrich, St. Louis, MI, USA) supplemented with 10% fetal bovine serum (Gibco), 100 U/mL penicillin, and 100 mg/mL streptomycin for 24 h at 37 °C in 5% CO_2_. For gene expression assessment, 0.3 × 10^6^ HEK293T cells/well were plated in a 6-well plate and cultured for 24 h until transfection. The plasmid DNAs, pVAX (empty vector), and pVAX-ScovSint1 were transfected in duplicate using a Lipofectamine™ 3000 Transfection Reagent kit (Invitrogen), following the manufacturer’s protocol. The cells were then incubated under the same conditions as previously described. After completion of the transfection step, the total RNA from cells was extracted using TRIzol™ Reagent (Invitrogen), and 1 µg of the total RNA obtained was treated with DNase I, RNase-free (Thermo Scientific, Waltham, MA, USA) and reverse-transcribed by a High-Capacity cDNA Reverse Transcription Kit (Thermo Scientific, Waltham, MA, USA) according to the manufacturers’ instructions. PCR was performed to confirm the in vitro expression of the multi-epitope synthetic antigen, ScovSint1. Primers were designed and synthesized to amplify a 145-base pair fragment representing the ScovSint1 synthetic gene (Primer Forward: ATGAAGAAGGGCACCAAC; Reverse: CCTTTGATCACCACGTTGG). PCR reactions were prepared with 1 µg of cDNA, and a 5× FIREPol^®^ Master Mix kit (Solis BioDyne, Tartu, Estonia) using Milli-Q nuclease-free water was used as a negative control. The resulting PCR products were observed after electrophoresis in 2% agarose gel.

### 2.8. Immunization Protocols and Characterization of Experimental Groups

The immunization schedule was performed in two doses, 1 week apart, with an intramuscular injection in the left tibialis anterior muscle (Figure 1). The vaccine administration was followed by in vivo electroporation with 8 pulses at 175 V/cm for 20 ms (BTX ECM 830, Harvard Apparatus). Before vaccine administration, all mice were anesthetized with xylazine hydrochloride (10 mg/Kg) and ketamine (115 mg/Kg). Twenty-one days after the first dose, all animals were euthanized.

The blood of all experimental animals was collected by cardiac puncture and stocked in EDTA tubes and tubes without anticoagulants. Besides the blood, the spleen was removed to perform the immunological assays. The spleen, heart, lung, liver, intestine, kidney, and a fragment of the paw muscle where the vaccines were inoculated were collected for histopathological analyses.

The experimental groups were (I) empty pVAX1 vector (negative control) and (II) Scov2-Sint1, each group composed of 10 mice. Female immunocompetent BALB/c mice, 6–8 weeks old, were raised and maintained in the biotherium of the Laboratório de Imunopatologia Keizo Asami (LIKA, Federal University of Pernambuco, Recife, PE, Brazil) under sterile, pathogen-free conditions. All experiments involving mice followed the standards established by the Institutional Ethics Committee for the Use of Animals (protocol no. 0054/2021).

### 2.9. Immunological Investigation

After euthanasia, mouse spleen cells were aseptically removed, isolated, and cultured following the protocol of Silva et al. 2021 [41]. Each spleen was placed in a tube containing supplemented RPMI 1640 (Sigma-Aldrich), and the cells were separated for maceration through a glass dounce. Cell suspensions were transferred to conical-bottom tubes containing Ficoll-Paque PLUS 1.077 g/mL (GE Healthcare) for the isolation of mononuclear cells. The cell suspensions were distributed in 24-well plates with 106 cells/well and cultured for 24 h in RPMI medium at 37 °C with 5% CO_2_.

Peripheral blood cells were isolated after erythrocyte lysis with hemolysis buffer (0.1 M sodium bicarbonate, 0.15 M ammonium chloride, and 0.1 mM EDTA, pH 7.2), followed by two washes with 1× PBS wash (10 min/1600 rpm). Cells were washed twice with PBS (400× *g*, 10 min) and labeled with anti-CD3-FITC, anti-CD4-PercyPCy5, anti-CD8-PE, anti-CD45-APC, anti-CD19-FITC, and anti-CD27-APC (BD Biosciences^®^, San Diego, CA, USA). The cell acquisition was performed with the same procedure applied to the blood cells. The cells were acquired in 10,000 events in a BD Accuri™ C6 Plus Flow Cytometer (BD Biosciences^®^), and data analysis was performed in the BD Accuri^®^ C6 software. The serum samples were used for cytokine dosage through a BD™ Cytometric Bead Array (CBA) Mouse Th1/Th2/Th17 CBA Kit (BD Bioscience^®^), and measured by flow cytometry.

### 2.10. Histopathological Investigation

Histological fragments of the liver, kidney, spleen, lung, intestine, heart, and muscles were obtained from previously anesthetized and euthanized mice. Tissue fragments were fixed in 10% formalin for 24 h. After fixation, the samples were washed with water and immersed in 70% ethyl alcohol for a period of 3 to 4 days, with subsequent dehydration of the material and replacement of the water present in the tissue by a dehydrating agent (ethanol) in increasing concentration (70%, 80%, 90%, 100%). The samples were cleared in Xylol and embedded in paraffin. The paraffin blocks were cut with a manual microtome (Leica, Wetzlar, Germany) to a thickness of 5 µm, stained with Hematoxylin and Eosin (HE), and mounted with coverslips using synthetic resin (Entellan, Merck). Histopathological analysis of tissue sections was performed using a microscopy system with the image capture software 2.3 ZEN blue edition with Axio Zeiss, NY, USA.

### 2.11. Statistical Analysis

The Student’s *t*-test was applied to assess statistical differences between control and the vaccine group. All results were expressed as the mean of the groups ± standard deviation values and analyzed considering the value of *p* < 0.05 as statistically significant. The statistical evaluation and graphs were processed in the GraphPad Prism 9 Software^®^.

## 3. Results

### 3.1. Selection and Analysis of T- and B-Cell Epitopes

A set of 11 epitopes consisting of 5 CTL epitopes and 6 HTL epitopes did not show any toxicity potential, and most of them had more than 80% conservation against emerging viral variants. The variant that most affected epitope conservation was Omicron (B.1.1.529) (Table 2). The epitope set showed 96.98% population coverage for combined class I and II HLAs worldwide (Figure 2) (Appendix A). Among the CTL epitopes, “LPFFSNVTW” was considered the most favorable in terms of stability with associated HLAs, because it presented the lowest free energy of −10.6 kcal/mol with HLA-B*35:01. The HTL epitope “VLSFELLHAPATVCG” presented −7.0 kcal/mol from docking with HLA-DRB1*01:01 (Figure 3).

Molecular dynamics simulations revealed that the seven complexes predicted in the molecular docking are stable under simulated conditions (Figure 4). The RMSD analysis from receptor-peptide atoms showed that all systems tend towards equilibrium after 30 ns (Figure 4A). The mean RMSD values of the complexes ranged from 1.94 to 4.38 Å, with an average of 3.05 Å. When considering only the atoms of the epitopes, all systems showed an equilibrium behavior from the middle of the simulation (Figure 4B). The average RMSD values of the peptides ranged from 1.19 to 4.55 Å, with an average of 2.55 Å. Notably, the HLA_B53-LPFFSNVTW, HLA_B35-LPFNDGVYF, HLA_A02-FQFCNDPFL, and HLA_C08-FQFCNDDPL complexes exhibited lower RMSD values compared with the other systems. The RMSF results (Figure 4C) showed that for all systems, the values per residue were close to 2 Å, with some regions exhibiting slightly higher variations. The HLA_DRB1-VLSFELLHAPATVC complex exhibited the greatest structural fluctuations, but since the variation was concentrated in the C-terminal region, and not in the cleft, it is unlikely to directly influence the interaction.

### 3.2. Multi-Epitope Design and Analysis

In the construction, the epitope were separated by a spacer sequence (linkers); for example, AAYs were added among the CTL epitopes, GPGPGs were added among the HTL epitopes, and KKs were added between the B-cell epitopes of the multi-epitope sequence. The spike protein epitopes were grouped and separated by a spacer sequence from the other set of nucleocapsid protein epitopes so that it is possible in future studies to digest these epitope groupings by separating them into two vaccine constructions, one with just spike epitopes and the other just with nucleocapsid. This way, it will be possible to make a comparison with the results of the construction of the present study (Figure 5A).

The multi-epitope antigen has 289 aa, 31,264.59 kDa, and its pI is 9.79, indicating a positive charge. The construct has 16 negatively charged residues (Asp + Glu) and 33 positively charged residues (Arg + Lys). Regarding this protein half-life, it is predicted to degrade by half after synthesis in 30 h in mammalian reticulocytes, in vitro; in yeasts, in vivo, it may take >20 h; and in *E. coli* cells, in vivo, it may take >10 h. The stability index of the molecule was 31.94, indicating that the protein is stable. The grand average of hydropathicity (GRAVY) was −0.461, and this showed that it has a hydrophilic nature, which facilitates interaction with water molecules, for example.

Furthermore, the multi-epitope construct was considered a probable antigen with a potential non-allergenic and soluble. Regarding the ability of the construction to cause cross-reactivity with the Homo sapiens genome, it presented a coverage of about 21% with 37.38% of identity (e-value = 9 × 10^−6^) for the spike protein and a coverage of about 33% for the nucleocapsid protein, with 37.04% of identity (e-value = 7 × 10^−8^) (Appendix A).

### 3.3. ScovSint1 Structure Modeling and Refinement

Five models of the 3D structure of ScovSint1 were predicted, with the first model selected, which presented a structure closest to the natural state. The differences between the structures before and after refinement were observed from the superposition (Figure 5B). The refined structure had its torsion angles improved compared with the unrefined one, as demonstrated by the Ramachandran plots, which in the refined model presented the amino acids positioned in a higher percentage of favorable regions (95%) (Figure 5C). In terms of secondary structure, this model has 4 sheets, 12 strands, 5 helices, 1 helix–helix interaction, 15 β-turns, and 11 γ-turns (Figure 5D).

### 3.4. Molecular Docking of ScovSint1 with TLRs and Immune Simulations

ClusPro generated approximately 30 clusters per docked complex, ScovSint1-TLR. Cluster 0 of each docking is considered the most stable, as it presents the lowest free energy of interaction between the ligand (ScovSint1) and the macromolecule (TLR). The free energy scores of cluster 0 related to ScovSint1 dockings with TLR2, TLR3, and TLR4 were −1090.8, −1036.7, and −1086.6 kcal/mol, respectively (Figure 6A) (Appendix A).

Regarding the stimulation of the immune response simulated by C-ImmSim, the designed vaccine antigen (ScovSint1) induces a strong cellular, humoral, and cytokine response after the second dose. This highlights the importance of the need for an administration regimen for this vaccine strategy that requires more than one dose to activate the immune response. When considering a homologous two-dose prime-boost administration regimen (the same antigen in both doses), with the first dose given on day 0 and the second dose on day 7, it was observed that the peaks of highest antigen concentrations occurred shortly after receiving the dose. From day 10 after receiving the first dose, it was observed that while their concentrations decreased, the IgM and IgG antibody titers increased over the following days, with their highest peaks between days 15 and 20 (Figure 6B). It was also observed that from the 10th day onwards, the T helper cells reached the peak of differentiation in a Th1 profile that assists in phagocytosis against intracellular microorganisms (Figure 6C). The increase in Th1 response results in the greater production of IFN-γ, IL-10, IL-2, IL-12, and TGF-B, as observed in Figure 6D, whose cytokine graph shows the highest peaks in this period. The levels of IFN-γ, in particular, stood out from the other cytokines by growing rapidly soon after the first dose, reaching the maximum peak of 5 × 105 ng/mL on the 10th day and beginning to decline on the 25th day onwards. More immune simulations can be observed in the Appendix A.

### 3.5. Cloning and In Vivo Vaccine Evaluation

#### 3.5.1. Obtaining the Expression Vector

The ScovSint1 sequence was cloned in silico into the pVAX1 vector (Figure 7A) and confirmed by enzymatic digestion (Figure 7B).

#### 3.5.2. Checking the Expression of the Vaccine Antigen, Scovsint1

The results confirmed the expression of the synthetic antigen ScovSint1 in transfected HEK-293T cells (Figure 7C). Conventional PCR analysis, using cDNA as a template, revealed a band corresponding to the amplified fragment of the ScovSint1 gene (145 bp), indicating successful transcription and cDNA synthesis from the RNA extracted from the cells. Additionally, no band was observed in the pVAX empty vector or in the reaction control, confirming the specificity of the amplification.

#### 3.5.3. Immunological Analysis

Analyses carried out 14 days after the second dose showed an increase in the number of CD4+ and CD8+ T lymphocytes in the blood (Figure 8A,B), with the induction of CD8+ memory cells (increased CD8+CD27+) (Figure 8B) and the secretion of IFN-γ (Figure 8C). There was no significant production of IL-2, IL-4, IL-6, IL-10, TNF-α, IFN-γ, and IL-17A (Appendix A). Evaluating the responses in the spleen cells, a stimulation of CD4+CD45+ and CD4+CD27+ cell production was observed (Figure 8D), followed by a slight increase in B cells (CD19+) (Figure 8F). The gating strategy to obtain these analyses is represented in the Appendix A.

### 3.6. Histopathological Analysis

The application of DNA vaccines did not cause damage in the vital organs, according to the histopathological analyses (Appendix A). The paw muscle of the group that received pVAX1 empty presented an area of necrosis and infiltration (Figure 9). These aspects can be commonly observed when this tissue receives a vaccine and undergoes procedures such as inoculation and electroporation.

## 4. Discussion

Here, the Spike and Nucleocapsid SARS-CoV-2 proteins were exploited to construct a synthetic antigen based on a multi-epitope vaccine. Immunoinformatics is an area that combines computational methods to analyze biological data associated with immune response. Several tools and online servers are developed to predict the most immunogenic regions of the pathogen and potential epitopes and analyze the conservation level in the face of variants, population coverage, anchoring and molecular dynamics with HLAs and immune receptors such as TLRs, allergenicity, antigenicity, induction of antibodies and cytokines, among other evaluations. Among the databases and tools that predict and analyze epitopes, the IEDB, used in the present study, is widely known for predicting epitopes from the pathogen protein sequence.

The recommended prediction method (NetMHCpan-4.1 and NetMHCIIpan-4.0) combines information regarding binding affinity, ligand elution, and length of the peptide that fits into the HLA cleft, besides other characteristics that are distributed in an artificial neural network. The peptides that fit all the characteristics associated with the strength of the interaction between the epitope and the corresponding HLA are considered epitopes. At the end, the tool ranks a list of epitopes based on a percentile score (the lower is the percentile rank, the greater is the epitope–HLA interaction) [42,43].

The 15 epitopes chosen to compose ScovSint1 were predicted from these recommended prediction methods mentioned. “LPFFSNVTW”, “LPFNDGVYF”, “FQFCNDPFL”, “QSLLIVNNATNVVIK”, “GNYNYLYRLFRKSNL”, “VLSFELLHAPATVCG”, “GTNGTKRFDN”, and “QSYGFQPTNGVGYQ” were also predicted by Rahman et al. [14], whereas “KMKDLSPRWYFY”, “DPNFKDQVILLN”, “ALLLLDRLNQLESKM”, “KAYNVTQAFGRRGPE”, and “FFGMSRIGMEVTPSG” were used in the study of Qamar et al. [44], and “EGALNTPKDHIGTRNP” and “KSAAEASKKPRQKRTA” were predicted by Dong et al. [6]. The aforementioned CTL and HTL epitopes were predicted and subsequently analyzed to identify their immunogenic potential, conservation, population coverage, and docking and molecular dynamics with HLAs.

As a multi-epitope vaccine, this strategy becomes more promising to stimulate the immune response, overcoming the low immunogenicity that DNA vaccines usually present [45]. In addition to the CTL and HTL epitopes, sequences referring to B-cell epitopes were also included in the multi-epitope. This aspect enhances the vaccine’s immune response, favoring a humoral response, which is important in the context of prophylaxis against COVID-19, relying on the production of neutralizing antibodies against the Spike and Nucleocapsid proteins of SARS-CoV-2.

After prediction, the epitopes underwent conservation, population coverage, docking, and molecular dynamics analyses of the epitope–HLA complex. Considering the effect of the variants of interest that emerged throughout the pandemic, the conservation analysis demonstrated the presence of some non-synonymous mutations that promoted amino acid substitutions, leading to exchange in the chemical groups, which can lead to changes in the peptide. As a result, alterations may occur in the molecule polarity and protein conformation or function [46]. The P.1 variant promoted an aspartic acid (D) to tyrosine (Y) exchange in the FQFCNDPFL epitope. This peptide is found in regions 131–141 of the Spike N-terminal portion [47]. Changes in this position can interfere with the conformation of RBD and the interactions with SARS-CoV-2 antibodies, impairing the neutralizing activity of antibodies produced by the vaccines manufactured during the pandemic beginning, before the identification of several mutations and variants [48,49]. Although new variants had appeared, the conservation percentage was high (80%).

Regarding the world population coverage, the epitope set of CTL and HTL reached 90%, similar to that obtained by Rahman et al. [14], who developed a multi-epitope vaccine with peptides of the SARS-CoV-2 structural proteins. The molecular docking analysis provided crucial structural insights into the interactions between epitopes and receptors. Moreover, the molecular dynamics simulations show that the HLA–epitope complexes remain stable under conditions similar to physiological ones. Notably, the RMSD values of the complexes are relatively low when compared with similar studies, indicating a high degree of stability [50,51,52].

The final sequence of the multi-epitope includes linkers between the epitopes to aid in the processing and presentation of the epitopes, such as AAY among the CTL epitopes that, after their cleavage, generate suitable sites for the transport of the epitopes via the transport of antigenic peptides (TAP) [28]; GPGPG among the HTL epitopes that stimulate responses mediated by CD4+ T lymphocytes [6,29,30]; and KK among the B-cell epitopes that are composed of lysine and target cathepsin B, the main favor for antigen presentation via MHC class II [31]. In the N-terminal region, a 7× HIS tag was added to facilitate recognition, from in vitro tests, of the synthesized multi-epitope protein; and in the C-terminal region, a TAT sequence was added that had the ability to improve nuclear delivery [53,54].

According to in silico analysis, the multi-epitope has a stable, soluble, antigenic, non-allergenic, non-toxic structure, and also does not have the ability to cause autoimmune response due to the low percentage of similarity with the *Homo sapiens* genome. The estimated half-life for mammalian reticulocytes in vitro is more than 30 h, similar to that in a study by Zhu et al. [55], and its grand average of hydropathicity (GRAVY) indicated a hydrophilic character, which avoids the need for micelles to carry this vaccine strategy in aqueous medium [55].

The model generated for the multi-epitope protein after refinement increased from 62.1% (unrefined) to 96% (refined), indicating an improvement in the quality of the model. Good-quality models usually have more than 90% of the residues in favorable regions [26,34]. The refined model of ScovSint1 was docked with different immune response receptors, as TLR2, TLR3, and TLR4. Among the docked complexes obtained, TLR2-ScovSint1 (−1090.8 kcal/mol) was the one with the lowest free energy, indicating greater stability. TLRs identify pathogen-derived products based on recognition patterns. TLR2, for example, recognizes membrane protein patterns [56].

The in silico simulation of the immune response generated by the multi-epitope vaccine antigen indicates the potential of the production of IgM and IgG antibodies and several cytokines, such as IFN-γ, TGF-β, IL-2, IL-10, and IL-12. Among these cytokines, IFN-γ and IL-2 stand out for reaching their highest peaks starting on the 10th day after the first dose. These results are similar to those reported by Cheng et al. [57], who also observed peaks of these cytokines, although they had adopted a longer experimental schedule than that used in the present study. It is noteworthy that these cytokines play crucial roles in the main immunological signals that mobilize the host defense against infections. IFN-γ is particularly significant for its involvement in T-cell-mediated antiviral responses, while IL-2 is essential for lymphocyte proliferation and the downregulation of viral replication [58,59].

Epitope-based vaccines have been developed for prevention and therapy against several infectious agents, including the Human Immunodeficiency Virus (HIV), *Plasmodium* sp., and *Mycobacterium tuberculosis* [60]. Despite numerous studies about the prediction and assembly of multi-epitope antigens, most of them do not include data from in vivo analysis that provide clues about the response generated by the vaccine antigen [6,7,8,9,10,11,51,60,61]. Here, we provide a preliminary in vivo evaluation of a synthetic antigen in a DNA vaccine approach.

Expression experiments using the HEK-293T cell model revealed amplification of the ScovSint1 gene fragment. This result demonstrates the efficiency of the transfection protocol, as the extracted RNA was successfully transcribed into cDNA and amplified by PCR. Furthermore, the absence of bands in the negative control reinforces the specificity of the results. These findings provide a solid foundation for future assays, such as evaluating protein expression by Western blotting, which will further validate the efficiency and stability of ScovSint1 expression in this system.

The natural infection promoted by SARS-CoV-2 elicits potent CD4+ and CD8+ T lymphocyte responses commonly associated with protective antiviral immunity [62]. Scov2-Sint1 promoted the increase in CD4+ and CD8+ T lymphocytes in the blood and in the spleen. In the analysis of the cellular response from spleen cells, the profile of CD4+ T cells stands out, with a significant increase in CD45+ and CD27+ cells. The result regarding the increase in CD4+/CD45+ cells is interesting, since this receptor acts in the T-cell regulation and activation [62,63]. On the other hand, signaling mediated via interactions with the CD27 receptor plays a role in T-cell-mediated responses, contributing to the activation and memory of T cells [64]. CD27 also influences CD8+ T-cell responses, increasing the survival of these cells and contributing to the generation of primary and memory cells in the context of viral infections [65,66]. An increase in the CD27 cell population was also observed in circulating CD8+ T cells.

Once the absence or low level of the SARS-CoV-2-specific T and B cells in the peripheral blood of patients with COVID-19 can result in greater disease severity, the results presented here are promising and indicate the potential protective response induced by the DNA multi-epitope vaccine [67]. SARS-CoV-2 can promote tissue damage in the spleen, such as loss of follicles in the white pulp and lymphoid hyperplasia. It was observed a reduction of the B-lymphocytes, CD3+ T-lymphocytes, and the Th2 cell number, and an increase in the Th1 and Th17 responses, in the spleen of patients with severe disease [67,68]. Remarkably, Scov2-Sint1 increased the number of effector and memory CD4+ T lymphocytes and B cells. Regarding the cytokine secretion, the synthetic antigen induced an increase in IFN-γ levels, as predicted in the in silico analysis.

Overall, the response induced by the multi-epitope vaccine denoted a cellular immune response profile similar to that achieved by other SARS-CoV-2 DNA vaccines [69,70]. Although the activation of B cells to generate specific neutralizing antibodies is fundamental in prophylactic immunization strategies, the T-lymphocyte-mediated response is also crucial to humoral response amplification and maintenance and for the promotion of long-term memory, and was the focus of the present study [71]. Furthermore, even in the face of emerging viral variants, the cellular immune response seems to be less affected than antibody-mediated responses [72]. Considering the spread of these variants and the fact that the majority of the world’s population has already been infected and/or vaccinated, it is important to think about vaccine strategies that contribute to sustaining protective immune responses.

## 5. Conclusions

Vaccine strategies that are easy to edit, such as nucleic acid vaccines associated with low-cost production platforms and rapid conservation verification of new variants, are exciting alternatives for the prophylaxis of diseases like COVID-19. It is worth noting that multi-epitope antigens require investment for in vitro and later in vivo validation to verify their immunization capacity in populations. Despite that most in silico vaccine studies often do not bring immunoinformatics data together with the in vivo validation, this work showed a complete panorama from the selection of epitopes to the induced immunological response. Additionally, it is important to consider that future investigations assessing the ability to stimulate the production of neutralizing antibodies and protective efficacy through viral challenge assays in preclinical models are crucial. Analyses of protein stability, potential toxicity, prolonged immunogenicity, and possible adverse effects would also be relevant to enhance the applicability and robustness of this study.

## Figures and Tables

**Figure 1 vaccines-13-00149-f001:**
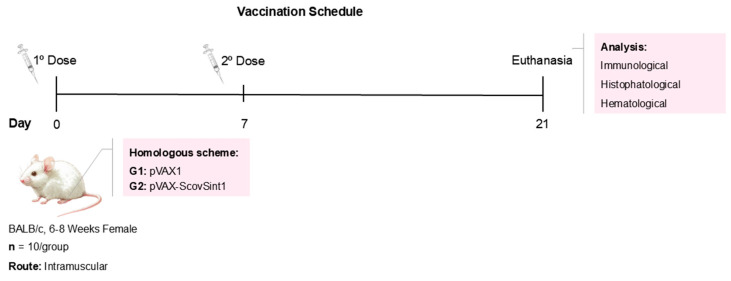
Vaccine scheme of the immunization schedule.

**Figure 2 vaccines-13-00149-f002:**
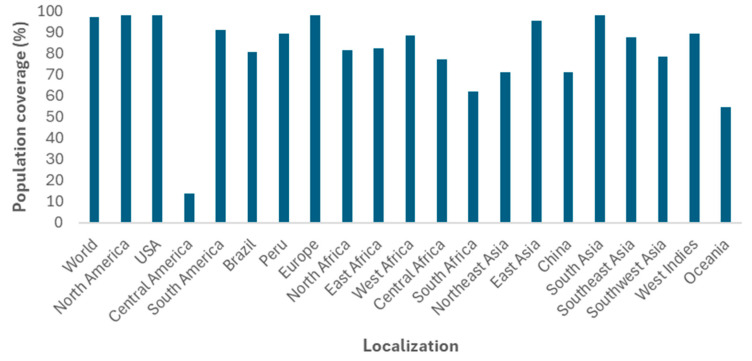
Population coverage of the synthetic antigen elaborated from the world allele frequencies, emphasizing the different continents and countries where the variants were found.

**Figure 3 vaccines-13-00149-f003:**
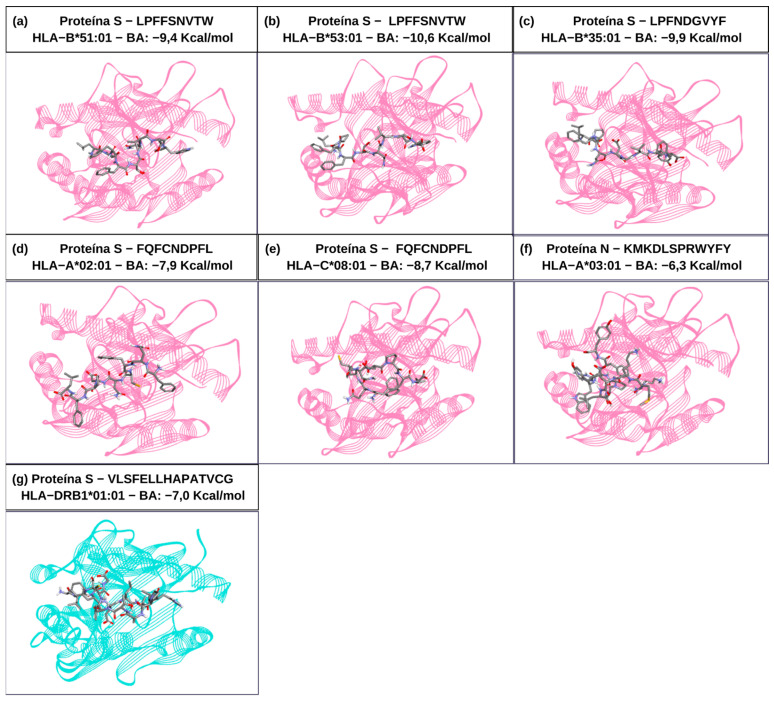
Epitopes of S and N proteins from the antigenic construct anchored through molecular docking with their respective class I (pink color) and class II (green color) HLAs and their binding affinity values (BA), corresponding to the binding energy between the epitope and HLA, with the lowest values having stronger binding affinity.

**Figure 4 vaccines-13-00149-f004:**
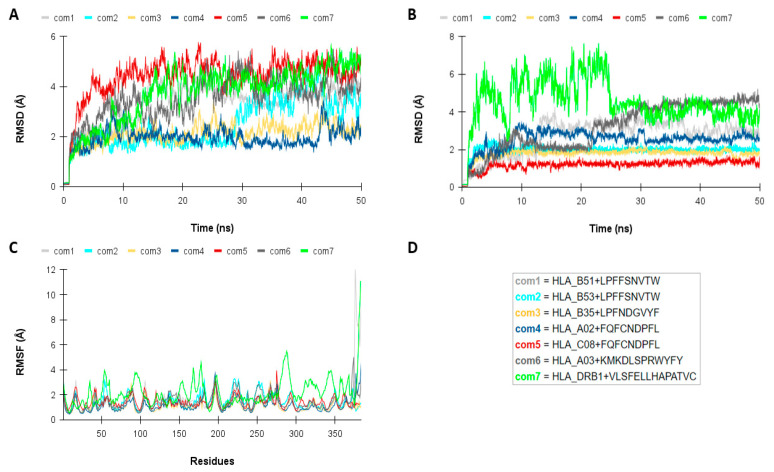
Graphics of root mean square deviation of atomic positions (RMSD) of molecular dynamics. (**A**) RMSD analysis of HLA–epitope complexes, comparing the structures at each step of simulation with their initial conformations, and considering the backbone atoms of both the receptor and the peptide. The analysis demonstrates that all complexes are stable. (**B**) RMSD analysis of epitopes, evaluating the predicted binding modes and showing that they tend to be maintained. (**C**) RMSF analysis of HLAs in each simulation, indicating that most residues exhibit low flexibility. (**D**) Abbreviations (“com” stands for “complex”).

**Figure 5 vaccines-13-00149-f005:**
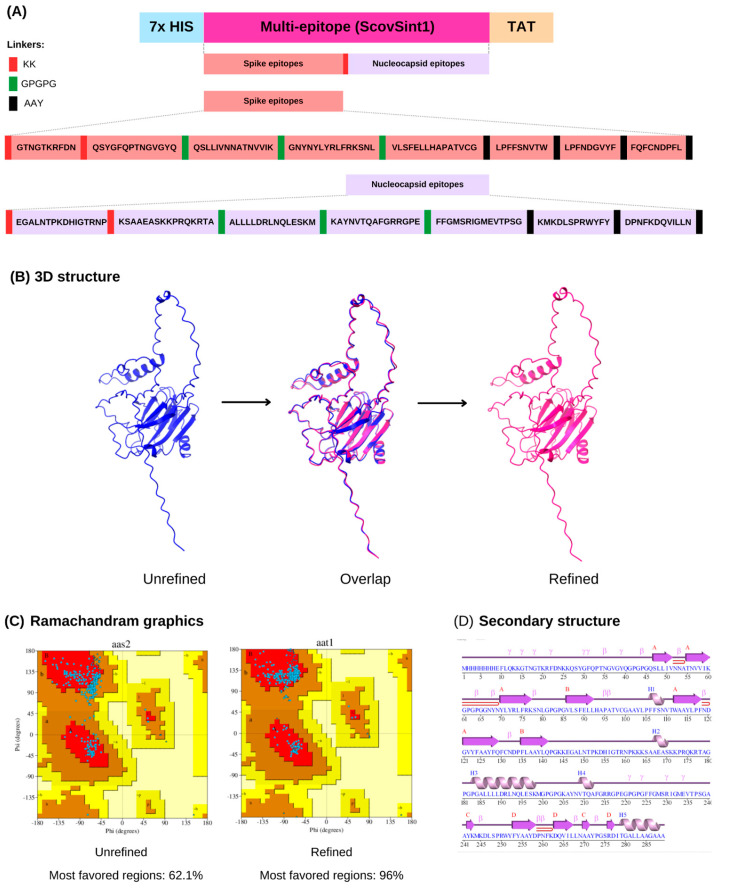
Schematic representation of virus structure, synthetic multi-epitope gene, and verification of the recombinant plasmid. (**A**) General scheme of the multi-epitope created by Illustrator for Biological Sequences (IBS) v1.0 (Version 1.0). KK, GPGPG, and AAY are linker sequences placed among the B-cell, HLA-I, and HLA-II epitopes. TAT sequence: cell-penetrating peptide that improves antigen delivery to the immune cells. (**B**) 3D structure of the multi-epitope protein ScovSint1 modeled before and after refinement. (**C**) Ramachandran plots showing the quality of the modeled protein before and after refinement, highlighting the position of the residues in the protein. Each residue is a point; the closer the point is to the red region, the better it is (>90% indicates a good quality model). (**D**) Secondary structure of the protein modeled, highlighting the composition, which is composed of sheets, strands, helices, helix–helix interaction, β-turns, and γ-turns.

**Figure 6 vaccines-13-00149-f006:**
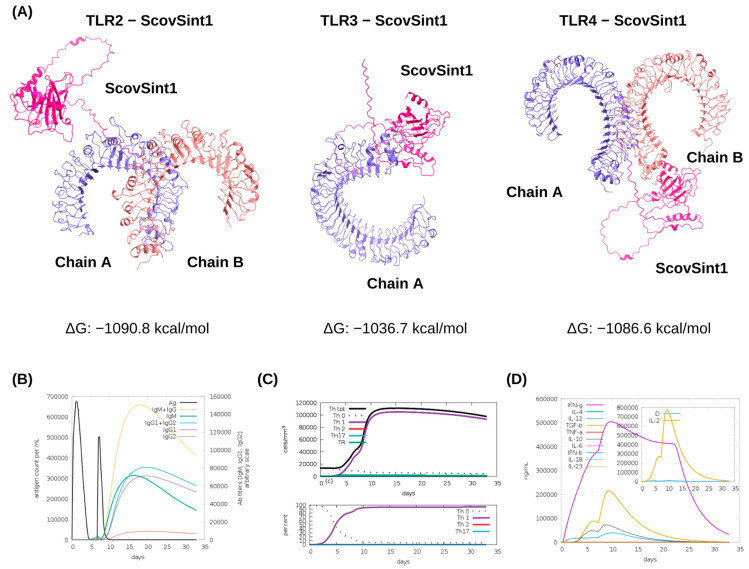
Representations of the docked models between the immune TLRs 2, 3, and 4 and ScovSint1 and simulations of the immune response throughout the immunization regimen (after 7 days—first dose and after 14 days—second dose). (**A**) ScovSint1-TLR2 was the model that presented the lowest free energy among three docked models with −1090.8 kcal/mol; (**B**) antibody production: all types of antibodies began production after the second dose; (**C**) the immune response profile: the generation of a Th1 response was indicated; (**D**) cytokine production: the cytokines produced in the greatest quantity were IFN-γ, TGF-B, IL-10, IL-12, and IL-2.

**Figure 7 vaccines-13-00149-f007:**
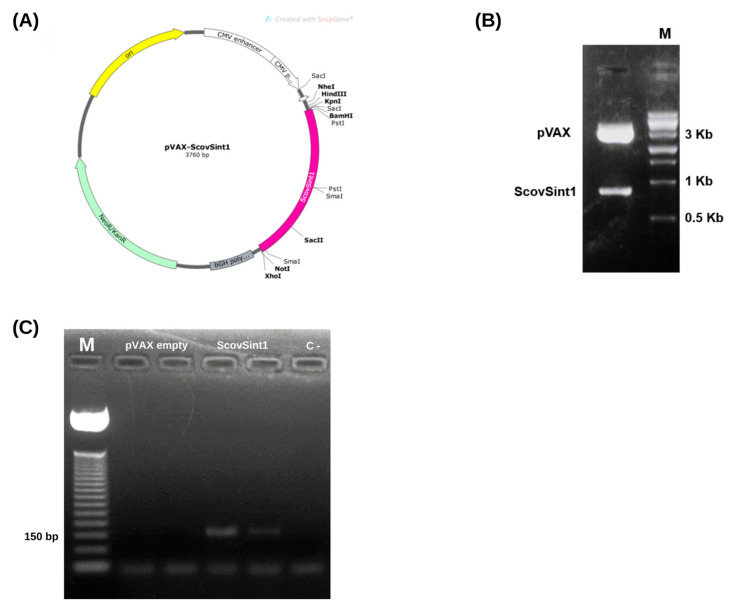
Cloning and expression analysis. (**A**) Schematic representation of the vector map. (**B**) Enzymatic cleavage (BamHI and NotI) of pVAX-ScovSint1 releasing the corresponding gene with 901 bp and the pVAX vector with 3000 bp. M: 1 Kb Ladder Sinapse Inc. (**C**) Amplification of Scovsint1 cDNA from RNA expressed in HEK-293T cells transfected with pVAX-ScovSint1 and empty pVAX as control confirming the expression of the corresponding gene (amplicon: 145 bp). M: 50 bp Ladder Promega.

**Figure 8 vaccines-13-00149-f008:**
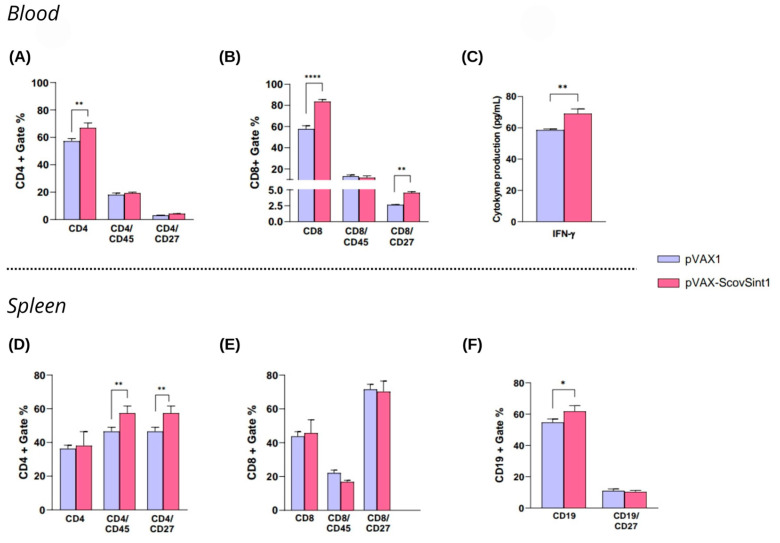
Profile of immune response induced by vaccination: (**A**–**C**) analysis of blood samples; (**D**–**F**) spleen cells’ evaluation. Asterisks represent statistical significance (* *p* < 0.05, ** *p* < 0.01, **** *p* < 0.0001). Bars indicate the mean value ± standard deviation.

**Figure 9 vaccines-13-00149-f009:**
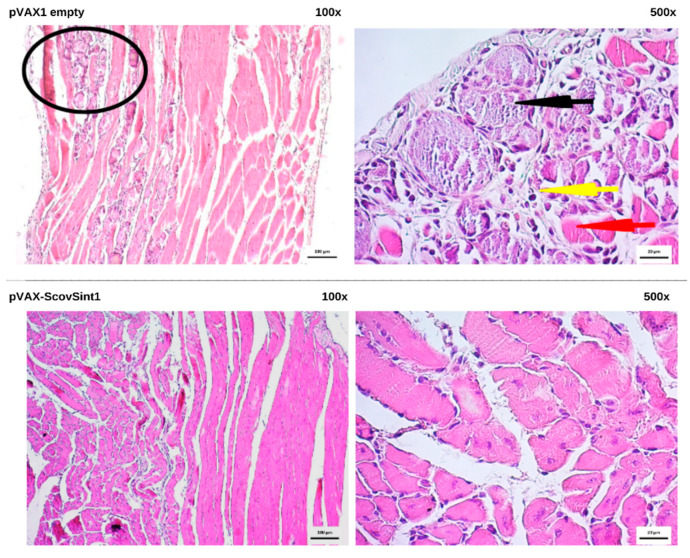
Representation of the histopathological samples of the paw muscle of pVAX1 empty (negative control) and pVAX-ScovSint1. In the skeletal muscle tissue of pVAX1 empty was an area of necrosis and infiltrate, while in the pVAX-ScovSint1 group, typical histologic features, absence of inflammatory infiltrate, or necrosis was observed. Description of symbols: red arrow: skeletal muscle fiber; black arrow: area of serous necrosis; yellow arrow: inflammatory infiltrate; black circle: area of necrosis and infiltrate.

**Table 1 vaccines-13-00149-t001:** HLAs used in the molecular docking of epitopes of vaccine construction.

HLA	Description	PDB ID	Structures Removed
Class I			
HLA-A*02:01	HLA-A*0201 human—ALWGPDPAAA.	3UTQ	Chain C: insulin
HLA-A*03:01	HIV RT-derived peptide in complex with HLA-A*0301.	3RL1	Chain C: peptide RT313
HLA-B*35:01	Insights into human allorecognition cross-reactivity: The structure of HLA-B35011 present in an epitope derived from cytochrome P450.	2CIK	Chain C: peptide Ligand: GOL (glycerol)



HLA-B*51:01	Binding of non-standard peptides from HLA-B*5101 complexed to the immunodominant epitope of HIV KM1 (LPPVVAKEI).	1E27	Chain C: HIV-1 peptide (LPPVVAKEI)



HLA-B*53:01	Class I MHC molecule B*5301 complexed with the LS6 peptide (KPIVQYDNF) from the malaria parasite P. falciparum.	1A1O	Chain C: LS6 peptide (KPIVQYDNF)


HLA-C*08:01	Crystal structure of HLA-C*0801.	4NT6	Chain C: matrix protein 1
Class II			
HLA-DRB1*01:01	Crystal structure of Staphylococcal Enterotoxin I (SEI) in complex with a human MHC class II molecule.	2G9H	Chain A: HLA class II histocompatibility antigen, DR alpha chain Chain C: hemagglutinin Chain D: extracellular type 1 enterotoxin Ligands: EPE (4-(2-hydroxyethyl)-1-piperazine ethanesulfonic acid), SO4 (sulfate ion), DIO (1,4-diethylene dioxide), and ZN (zinc ion)










**Table 2 vaccines-13-00149-t002:** Sequences and correspondent HLA of S and N epitopes selected for the construction of the multi-epitope antigen.

Epitope *	Sequence	Receptor	HLA	Percentile Rank	Immunogenicity	Conservation (%)	Mutation (Variant)
S_56–64_	LPFFSNVTW	HLA-I	HLA-B*53:01	0.01	0.04613	100 (27/27)	
			HLA-B*51:01	0.2			
S_84–92_	LPFNDGVYF	HLA-I	HLA-B*35:01	0.01	0.11767	96 (26/27)	**NS**F**TR**GVY**Y (Omicron—B.1.1.529)**
			HLA-B*15:02	0.12			
S_131–141_	FQFCNDPFL	HLA-I	HLA-A*02:06	0.17	0.05737	96 (26/27)	FQFCN**Y**PFL **(Brazil—P.1)**
			HLA-C*08:01	0.56			
			HLA-A*02:01	0.41			
S_115–129_	QSLLIVNNATNVVIK	HLA-II	HLA-DRB1*13:02	1.9	-	100 (27/27)	
S_447–461_	GNYNYLYRLFRKSNL	HLA-II	HLA-DRB1*11:01	4.5	-	80 (22/27)	GNYNY**R**YRLFRKSNL **(USA—B.1.427 and B.1.429** **India—B.1.617.1 and B.1.617.2** **Peru—C.37)**
S_512–526_	VLSFELLHAPATVCG	HLA-II	HLA-DRB1*01:01	0.67	-	100 (27/27)	
S_72–81_	GTNGTKRFDN	B cell	-		-	92 (25/27)	GTNGTKRF**A**N **(South Africa—B.1.351)** GTN**VI**KRFDN **(Peru—C.37)**
S_493-506_	QSYGFQPTNGVGYQ	B cell	-		-	85 (23/27)	QSYGFQPT**Y**GVGYQ **(United Kingdom—B.1.1.7** **South Africa—B.1.351** **Brazil—P.1)** **YQA**G**NK**P**C**NGV**AGF** **(Omicron—B.1.1.529)**
N_100–111_	KMKDLSPRWYFY	HLA-I	HLA-A*29:02	0.97	0.09381	96 (26/27)	**NVSLVK**P**SF**Y**V**Y **Omicron (B.1.1.529)**
			HLA-E*01:01	13			
			HLA-A*32:01	1.5			
			HLA-A*01:01	3.1			
			HLA-A*03:01	2.6			
			HLA-A*31:01	3.9			
			HLA-B*15:01	0.59			
N_343–354_	DPNFKDQVILLN	HLA-I	HLA-B*35:03	12	−0.03586	96 (26/27)	**SFYVYSRVKN**LN **Omicron (B.1.1.529)**
N_220–234_	ALLLLDRLNQLESKM	HLA-II	HLA-DRB1*11:04	17	-	88 (24/27)	ALLLLDRLNQLESK**I** **(Brazil—P.2** **USA—B.1.526)** AL**R**L**CAYCC**N**IVNVS (Omicron—B.1.1.529)**
			HLA-DRB1*11:06	20			
			HLA-DRB1*13:11	17			
			HLA-DRB1*13:21	38			
			HLA-DRB1*13:07	40			
			HLA-DRB1*11:02	39			
			HLA-DRB1*11:21	40			
			HLA-DRB1*13:22	39			
			HLA-DRB1*13:04	32			
			HLA-DRB1*08:06	17			
			HLA-DRB1*11:28	24			
			HLA-DRB1*13:05	24			
			HLA-DRB1*08:04	32			
			HLA-DRB1*11:14	40			
			HLA-DRB1*13:23	40			
N_266–280_	KAYNVTQAFGRRGPE	HLA-II	HLA-DRB5*01:05	0.03	-	96 (26/27)	**C**AY**CCNIVNVSLVKP** **(Omicron—B.1.1.529)**
			HLA-DRB5*01:01	0.03			
N_316–330_	FFGMSRIGMEVTPSG	HLA-II	HLA-DRB1*11:06	61	-	96 (26/27)	F**YVY**SR**VKNLNSSRV (Omicron—B.1.1.529)**
			HLA-DRB1*08:04	63			
			HLA-DRB1*11:28	63			
			HLA-DRB1*13:05	63			
			HLA-DRB1*13:21	71			
N_138–153_	EGALNTPKDHIGTRNP	B cell			-	92 (25/27)	EGALNTPKDHIG**I**RNP **China-Shenzhen (Omicron—B.1.1.529)**
N_251–266_	KSAAEASKKPRQKRTA	B cell			-	96 (26/27)	**NIVNVSLV**KP**SFYVYS** **(Omicron—B.1.1.529)**

* Note: All epitopes are non-toxic.

## Data Availability

All relevant data from this study are available from the corresponding author.

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
