# Peer review of "Design and Immune Profile of Multi-Epitope Synthetic Antigen Vaccine Against SARS-CoV-2: An In Silico and In Vivo Approach"

_vaccines, 2025, doi:10.3390/vaccines13020149_

Round 1

Reviewer 1 Report

Comments and Suggestions for Authors

Maria et al. have developed a multiepitope vaccine against SARS-CoV-2 using in silico and in vivo approaches. The epitopes focus on immunogenic regions of the spike and nucleocapsid proteins, utilizing a DNA vaccine-based platform. Interestingly, the developed vaccine induces CD4+ and CD8+ cells and shows no tissue damage in histopathological analysis. Overall, the study is intriguing; however, several questions need to be addressed:

  1. Did the authors perform a virus neutralization assay?

  2. Why did the authors not perform challenge studies?

  3. Why were the immunization studies designed with the second dose given on day 7 instead of day 14?

  4. The FACS analysis lacks clarity. The spleen isolation methodology is unclear, and acquiring only 10,000 cells seems insufficient to draw conclusions. Moreover, the authors only showed CD4+ and CD8+ populations without staining for other cytokine populations or effector and central memory cells.

  5. Did the authors review the histology slides blindly, with input from a pathologist?

Author Response

REBUTTAL LETTER

(Round 1)

Recife, December 14, 2024

Dear Editor,

I am submitting the revised version of article entitled “Design And Immune Profile of Multi-Epitope Synthetic Antigen Vaccine Against SARS-CoV-2: an in silico and in vivo Approach” by Invenção et al., for publication in the Vaccines, as part of the Special Issue entitled "New Approaches to Vaccine Development and Delivery".

All modifications are highlighted in the manuscript, and the response to reviewers follows below. All the authors confirm that they saw and agreed to the submitted paper. The authors have been recognized as contributors and have agreed to their inclusion. The material is original, and it has been neither published elsewhere nor submitted for publication simultaneously. None of the authors has any potential financial conflict of interest related to this manuscript.

 We appreciate the suggestions. The highlighted text has been revised.

Managing Editor 

Please revise during the revision:

  1. Structure Abstract (background/methods/results/conclusions) is needed,

here is an example: https://www.mdpi.com/2076-393X/12/12/1343

  1. Reduce the similarity during revision.
  2. Sign the authorship statement (in attachment), this can be done before we

send it to a final decision.

Response: We appreciate the points highlighted and have made the requested corrections.

Reviewer 1

Maria et al. have developed a multiepitope vaccine against SARS-CoV-2 using in silico and in vivo approaches. The epitopes focus on immunogenic regions of the spike and nucleocapsid proteins, utilizing a DNA vaccine-based platform. Interestingly, the developed vaccine induces CD4+ and CD8+ cells and shows no tissue damage in histopathological analysis. Overall, the study is intriguing; however, several questions need to be addressed:

  1. Did the authors perform a virus neutralization assay? Response: We appreciate the suggestion regarding virus neutralization assays, recognizing their importance as one of the main parameters for the evaluation of prophylactic vaccines. However, the main focus of the submitted manuscript was the construction and characterization of a synthetic multi-epitope antigen of SARS-CoV-2 and the demonstration of its immunogenicity as a DNA vaccine. The assays performed focused on the evaluation of the cellular immune response induced by the antigen, evidencing the expansion of CD4+ and CD8+ T cells, as well as on the histopathological evaluation correlated with the vaccine. Thus, the results obtained so far provide a solid basis for future studies, including virus neutralization assays, which could complement the evaluation of the prophylactic potential of the vaccine.
  2. Why did the authors not perform challenge studies? Response: We agree with the importance of including animal challenges to assess the induction of protection. Nonetheless, the experimental design adopted in the submitted article is similar to that of other studies that aimed to evaluate vaccine candidates against SARS-CoV-2 infection. Even without performing the immunological challenge, our study points out the immunogenic potential of recombinant yeasts, showing a response pattern close to the expected in the context of natural SARS-CoV-2 infection and through vaccination (Gao et al., 2021; Xing et al., 2022; Liu et al., 2022; Liu et al., 2024; Pollet et al., 2021).

Gao T, Ren Y, Li S, Lu X, Lei H. Immune response induced by oral administration with a Saccharomyces cerevisiae-based SARS-CoV-2 vaccine in mice. Microb Cell Fact. 2021 May 5;20(1):95. doi: 10.1186/s12934-021-01584-5.

Xing H, Zhu L, Wang P, Zhao G, Zhou Z, Yang Y, Zou H, Yan X. Display of receptor-binding domain of SARS-CoV-2 Spike protein variants on the Saccharomyces cerevisiae cell surface. Front Immunol. 2022 Aug 12;13:935573. doi: 10.3389/fimmu.2022.935573.

Liu Y, Zhao D, Wang Y, Chen Z, Yang L, Li W, Gong Y, Gan C, Tang J, Zhang T, Tang D, Dong X, Yang Q, Valencia CA, Dai L, Qi S, Dong B, Chow HY, Li Y. A vaccine based on the yeast-expressed receptor-binding domain (RBD) elicits broad immune responses against SARS-CoV-2 variants. Front Immunol. 2022 Nov 9;13:1011484. doi: 10.3389/fimmu.2022.1011484.

Liu Y, Li M, Cui T, Chen Z, Xu L, Li W, Peng Q, Li X, Zhao D, Valencia CA, Dong B, Wang Z, Chow HY, Li Y. A superior heterologous prime-boost vaccination strategy against COVID-19: A bivalent vaccine based on yeast-derived RBD proteins followed by a heterologous vaccine. J Med Virol. 2024 Mar;96(3):e29454. doi: 10.1002/jmv.29454.

Pollet J, Chen WH, Versteeg L, Keegan B, Zhan B, Wei J, Liu Z, Lee J, Kundu R, Adhikari R, Poveda C, Villar MJ, de Araujo Leao AC, Altieri Rivera J, Momin Z, Gillespie PM, Kimata JT, Strych U, Hotez PJ, Bottazzi ME. SARS‑CoV-2 RBD219-N1C1: A yeast-expressed SARS-CoV-2 recombinant receptor-binding domain candidate vaccine stimulates virus neutralizing antibodies and T-cell immunity in mice. Hum Vaccin Immunother. 2021 Aug 3;17(8):2356-2366. doi: 10.1080/21645515.2021.1901545. 

3. Why were the immunization studies designed with the second dose given on day 7 instead of day 14? Response: The vaccination schedule with the second dose administered on day 7 was adopted based on protocols previously validated by our research group, as described by Silva et al. (2023). Although the 14-day interval is also widely used in immunization studies, its application may vary depending on the type of antigen, the experimental model, and the specific objectives of the study. For this research, the decision to use the 7-day interval was based on the need to evaluate the initial efficacy of the multi-epitope antigen in inducing rapid immune responses. We emphasize, however, that for this specific research, we have not yet conducted comparative studies with alternative vaccination schedules. These future tests may contribute to further optimizing the interval between doses, adjusting it to the characteristics of the antigen developed and the desired immunological profile.

4. The FACS analysis lacks clarity. The spleen isolation methodology is unclear, and acquiring only 10,000 cells seems insufficient to draw conclusions. Moreover, the authors only showed CD4+ and CD8+ populations without staining for other cytokine populations or effector and central memory cells. Response: The description of the methodology for splenocyte isolation was corrected and detailed in the manuscript. The cytometer settings were set to 10,000 events, as indicated for immunological analyses by cytometry, which is a standard setting and collects sufficient cells for analysis (Cossarizza et al. 2021). Not only the labeling for CD4+ and CD8+ cells was shown, since Figure 8 also shows the profile of CD45+ and CD27+ cells. The discussion regarding these important markers for activation and generation of memory in T cells was added to the text. Figure 8 also shows that IFN-Y was detected in the serum of the animals, in line with the increase in CD4+ and CD8+ T cells, suggesting the induction of a Th1 profile.

Cossarizza, A.; Chang, H.; Radbruch, A.; Akdis, M.; Andrä, I.; Annunziato, F.; Bacher, P.; Barnaba, V.; Battistini, L.; Bauer, W.M.; et al. Guidelines for the Use of Flow Cytometry and Cell Sorting in Immunological Studies*. Eur J Immunol 2017, 47, 1584–1797, doi:10.1002/eji.201646632.

5. Did the authors review the histology slides blindly, with input from a pathologist? Response: The histology slides were prepared and read by co-author Julliano Matheus de Lima Maux, a biomedical scientist and clinical pathologist, who had experience working with anatomopathological studies in human and murine models. In addition, they were reviewed by co-author Prof. Dr. Jacinto Costa, who is a renowned cytohistopathologist in Brazil. In addition to reviewing the slides, he was also the one who captured the histological images for inclusion in this study.

Reviewer 2 Report

Comments and Suggestions for Authors

The authors analyzed the structure of the spike and nucleocapsid epitopes of SARS-CoV-2 and incorporated them into the pVAX1 vector as a DNA vaccine. A significant flaw of this study is that it did not verify the situation of the DNA vaccine in vivo, such as expression. It remains unproven that the DNA vaccine was effective in vivo.

Major:

1.    Following the predictions from Table 2, Figure 2, Figure 3, and Figure 4 generated by the software, the authors need to conduct additional experiments to validate their predictions. For instance, capillary electrophoresis should be employed to verify the affinity of the S protein peptide with HLA.

2.    Why was there no control for inserting a single spike epitope and a single nucleocapsid epitope into the pVAX1 vector? This control is necessary to compare the advantages of ScovSint1.

3.    There is limited comparability between the predicted experimental results and the actual experimental outcomes. For example, the predicted immune response indicates that ScovSint1 can induce high levels of IgM, IgG1, and Th1. However, why were these antibody or cell levels not detected in the actual experiment? These data should support the reliability of the predicted results.

4.    The predicted immune response suggested that ScovSint1 could induce high levels of IFN-γ, TGF-β, IL-2, and other cytokines. However, the actual experiment showed a significant difference only for IFN-γ, while IL-2, which was predicted to be at a higher level than IFN-γ, showed no significant difference. Does this indicate that the modeling of the predicted immune response diverges significantly from actual conditions and is therefore unreliable?

Minor:

1.    Not all data need to be included in the text. For example, the toxicity column does not need to be listed separately in Table 2 but can be mentioned in the table notes.

2.    In Table 1, the format is misaligned. For instance, is the content for 3RL1 in the PDB ID column aligned with HLA-A03:01 and chain C: peptide RT313? Does the description "Binding of non-standard peptides from HLA-B5101 complexed to the immunodominant epitope of HIV KM1 (LPPVVAKEI)" align with HLA-B*51:01, 1E27, and chain C: HIV-1 peptide (LPPVVAKEI)? Additionally, the upper part of 1A1O in the PDB ID column is not aligned, and the row corresponding to chain C: matrix protein 1 in the Structures removed column is also misaligned.

3.    In Table 2, the header font is incomplete and the format is misaligned. For example, is the content GTNGTKRFAN (South Africa B.1.351) in the mutation column aligned with S72-81? Are -0.03586 in the Immunogenicity column, Non-toxic in the Toxicity column, and 96 (26/27) in the Conservation column aligned with N343-354?

4.    There are inconsistencies between the text in the article and the figures. For instance, lines 329-330 and 343 refer to "IFN-γ" in the article, but "IFN-g" is used in the figure.

Author Response

REBUTTAL LETTER

(Round 1)

Recife, December 14, 2024

Dear Editor,

I am submitting the revised version of article entitled “Design And Immune Profile of Multi-Epitope Synthetic Antigen Vaccine Against SARS-CoV-2: an in silico and in vivo Approach” by Invenção et al., for publication in the Vaccines, as part of the Special Issue entitled "New Approaches to Vaccine Development and Delivery".

All modifications are highlighted in the manuscript, and the response to reviewers follows below. All the authors confirm that they saw and agreed to the submitted paper. The authors have been recognized as contributors and have agreed to their inclusion. The material is original, and it has been neither published elsewhere nor submitted for publication simultaneously. None of the authors has any potential financial conflict of interest related to this manuscript.

 We appreciate the suggestions. The highlighted text has been revised.

Managing Editor 

Please revise during the revision:

  1. Structure Abstract (background/methods/results/conclusions) is needed,

here is an example: https://www.mdpi.com/2076-393X/12/12/1343

  1. Reduce the similarity during revision.
  2. Sign the authorship statement (in attachment), this can be done before we

send it to a final decision.

Response: We appreciate the points highlighted and have made the requested corrections.

Reviewer 2

The authors analyzed the structure of the spike and nucleocapsid epitopes of SARS-CoV-2 and incorporated them into the pVAX1 vector as a DNA vaccine. A significant flaw of this study is that it did not verify the situation of the DNA vaccine in vivo, such as expression. It remains unproven that the DNA vaccine was effective in vivo. Response: We appreciate the suggestion and acknowledge the importance of these results for the research. In the context of this study, the expression analysis was performed using the HEK-293T cell model, as added in the methods and results sections. The amplification of the ScovSint1 gene fragment (145 bp) was observed, confirming the efficiency of the transfection protocol.. As indicated in the discussion, these findings provide a solid foundation for future assays, such as the assessment of protein expression by Western blotting, which will further validate the efficiency and stability of ScovSint1 expression in this system.

Major:

  1.   Following the predictions from Table 2, Figure 2, Figure 3, and Figure 4 generated by the software, the authors need to conduct additional experiments to validate their predictions. For instance, capillary electrophoresis should be employed to verify the affinity of the S protein peptide with HLA. Response: We appreciate your comments, but capillary electrophoresis would only separate these molecules based on size without providing information that validates the binding affinity between epitope and HLA that were predicted by Immunoinformatics tools. Thus, the implementation of this technique to validate these results is not implemented in published articles on multi-epitope such as, for example,

Guo, N.; Niu, Z.; Yan, Z.; Liu, W.; Shi, L.; Li, C.; Yao, Y.; Shi, L. Immunoinformatics Design and In Vivo Immunogenicity Evaluation of a Conserved CTL Multi-Epitope Vaccine Targeting HPV16 E5, E6, and E7 Proteins. Vaccines 2024, 12, 392. https://doi.org/10.3390/vaccines12040392 

Kaushik V, G SK, Gupta LR, Kalra U, Shaikh AR, Cavallo L and Chawla M (2022) Immunoinformatics Aided Design and In-Vivo Validation of a Cross-Reactive Peptide Based Multi-Epitope Vaccine Targeting Multiple Serotypes of Dengue Virus. Front. Immunol. 13:865180. doi: https://doi.org/10.3389/fimmu.2022.865180 

Ardestani, H., Nazarian, S., Hajizadeh, A., Sadeghi, D., & Kordbacheh, E. (2022). In silico and in vivo approaches to recombinant multi-epitope immunogen of GroEL provides efficient cross protection against S. Typhimurium, S. flexneri, and S. dysenteriae. Molecular Immunology, 144, 96-105. https://doi.org/10.1016/j.molimm.2022.02.013 

  1.   Why was there no control for inserting a single spike epitope and a single nucleocapsid epitope into the pVAX1 vector? This control is necessary to compare the advantages of ScovSint1. Response: The focus of this study was to characterize the immune response induced by the full sequence of the synthetic antigen, which combines immunogenic regions of the SARS-CoV-2 Spike and Nucleocapsid proteins. In the in silico evaluation, we performed the analyses for the full multiepitope sequence, and thus, the in vivo analyses are in agreement. Different studies present information on the response of the Spike protein as well as the Nucleocapsid, and our idea was to evaluate a construct that included immunogenic epitopes of both proteins.

  1. There is limited comparability between the predicted experimental results and the actual experimental outcomes. For example, the predicted immune response indicates that ScovSint1 can induce high levels of IgM, IgG1, and Th1. However, why were these antibody or cell levels not detected in the actual experiment? These data should support the reliability of the predicted results. Response: We appreciate your comments, but these data of immune response about antibodies and Th1 in vivo was not included because it was not made. We appreciate the suggestion. The synthetic antigen tested is composed mainly of epitopes targeting T cells, which favors a more prominent cellular immune response profile. Since our study focuses on in silico analysis and an initial assessment to understand this type of response, the experiments performed do not yet include an in-depth analysis of the humoral profile. Based on the results obtained, we intend to expand future studies to evaluate, in detail, the humoral response and other relevant immunological aspects, in order to better correlate the predicted data with the actual experimental data.

  1.   The predicted immune response suggested that ScovSint1 could induce high levels of IFN-γ, TGF-β, IL-2, and other cytokines. However, the actual experiment showed a significant difference only for IFN-γ, while IL-2, which was predicted to be at a higher level than IFN-γ, showed no significant difference. Does this indicate that the modeling of the predicted immune response diverges significantly from actual conditions and is therefore unreliable? Response: We appreciate your comments, but it is important to understand that the C-IMMSIM server performed the prediction in which the presence of these cytokines and immune response profile were identified in silico for periods up to (35 days) the period of the immunization regimen performed in this study (21 days in total, with euthanasia for organ removal for immunological analyses of cytokines and cellular response being performed on the last day). Thus, the period in which the immunization regimen was performed may not have detected the other cytokines in silico predicted for the following days. Furthermore, as shown in the papers on multi-epitope vaccines published below (which represent the majority of papers in this context), in vivo data are not usually included in these in silico papers, so that the bioinformatics tools seek to predict the experimental results, but do not guarantee total fidelity of the results in the experimental pratice, thus requiring validation in subsequent in vivo studies.  Thus, this manuscript presents a differential aspect, since in addition to predicting, it sought to experimentally validate the predicted data and thus showed that despite the match data (e.g.: IFN-Y induction), there are also data in the experimental practice that differ from what had been predicted. Considering these scenarios, the observed results do not invalidate their confidence.

Sanami, S., Rafieian-Kopaei, M., Dehkordi, K.A. et al. In silico design of a multi-epitope vaccine against HPV16/18. BMC Bioinformatics 23, 311 (2022). https://doi.org/10.1186/s12859-022-04784-x 

Maleki, A., Russo, G., Parasiliti Palumbo, G.A. et al. In silico design of recombinant multi-epitope vaccine against influenza A virus. BMC Bioinformatics 22 (Suppl 14), 617 (2021). https://doi.org/10.1186/s12859-022-04581-6 

Saha, S., Vashishtha, S., Kundu, B. et al. In-silico design of an immunoinformatics based multi-epitope vaccine against Leishmania donovani. BMC Bioinformatics 23, 319 (2022). https://doi.org/10.1186/s12859-022-04816-6 

Kumar, A., Sahu, U., Agnihotri, G., Dixit, A., & Khare, P. (2024). A novel multi‐epitope peptide vaccine candidate targeting hepatitis E virus: An in silico approach. Journal of Viral Hepatitis, 31(8), 446-456. https://doi.org/10.1111/jvh.13949 

Kafle, A., Tenorio, J. C. B., Mahato, R. K., Dhakal, S., Heikal, M. F., & Suttiprapa, S. (2024). Construction and validation of a novel multi-epitope in silico vaccine design against the paramyosin protein of Opisthorchis viverrini using immunoinformatics analyses. Acta Tropica, 260, 107389. https://doi.org/10.1016/j.actatropica.2024.107389 

Minor:

  1.   Not all data need to be included in the text. For example, the toxicity column does not need to be listed separately in Table 2 but can be mentioned in the table notes. Response: We appreciate your considerations. We have made the suggested changes.

  1.   In Table 1, the format is misaligned. For instance, is the content for 3RL1 in the PDB ID column aligned with HLA-A03:01 and chain C: peptide RT313? Does the description "Binding of non-standard peptides from HLA-B5101 complexed to the immunodominant epitope of HIV KM1 (LPPVVAKEI)" align with HLA-B*51:01, 1E27, and chain C: HIV-1 peptide (LPPVVAKEI)? Additionally, the upper part of 1A1O in the PDB ID column is not aligned, and the row corresponding to chain C: matrix protein 1 in the Structures removed column is also misaligned. Response: We appreciate your considerations. We have made the suggested changes.

  1.   In Table 2, the header font is incomplete and the format is misaligned. For example, is the content GTNGTKRFAN (South Africa – B.1.351) in the mutation column aligned with S72-81? Are -0.03586 in the Immunogenicity column, Non-toxic in the Toxicity column, and 96 (26/27) in the Conservation column aligned with N343-354? Response: We appreciate your considerations. We have made the suggested changes.

  1. There are inconsistencies between the text in the article and the figures. For instance, lines 329-330 and 343 refer to "IFN-γ" in the article, but "IFN-g" is used in the figure. Response: We appreciate your considerations, and we have standardized the possible terms throughout the article. In this exemplified situation of IFN-γ, the graphs generated by C-IMMSIM already come standardized with the cytokine legends as presented in the work, so we cannot change this detail without losing the resolution of the figure.

Reviewer 3 Report

Comments and Suggestions for Authors

Reviewer comments

vaccines-3337996

“Design And Immune Profile of Multi-Epitope Synthetic Antigen Vaccine Against SARS-CoV-2: an in silico and in vivo Approach”

This study needs MAJOR revision.

1.     The study is interesting, but the writing is not clear and may cause confusion for readers. The overall paper should be revised to make it clear for readers.

2.     Highlight the unique aspects of this study compared to existing other vaccine developing strategies.

3.     In the abstract. The statement “The Spike and Nucleocapsid epitopes, 25 described in previous prediction studies, were organized in a database, and verified for 26 immunogenicity, conservation, population coverage, and molecular docking values” is not clear.

4.     Line 35. Remove. “therefore”

5.     Only 3 keywords.

6.     Line 52. “COVID-19 vaccines must be able to elicit especially Th1 response (IL-2, IL-6, TNF, 52 and IFN), with an increase of CD8 T cell action, the CD4 T and B memory cells, and 53 promoting the anti-SARS-CoV-2 NAb long term production.” Link this statement properly.

7.     Line 64. This sentence is ambiguous “It is possible to select epitopes that contribute to increased efficacy and immunogenicity 64 concerning vaccines under development through accurate data, requiring low-cost tools 65 once the analyzes can be performed through free software.”

8.     Section 2.2. can be make short and some information may be kept in supplementary materials.

9.     Check Abbreviation and Full names throughout the paper. For example, line 320 and many more….

10.  Lines 346-348, 394-396, 470-471, 548-550, 560-565, . Check for clarity.

Round 2

Reviewer 1 Report

Comments and Suggestions for Authors

Accept in present form

Author Response

REBUTTAL LETTER

(Round 2)

Recife, December 19, 2024

Dear Editor,

I am submitting the revised version of article entitled “Design And Immune Profile of Multi-Epitope Synthetic Antigen Vaccine Against SARS-CoV-2: an in silico and in vivo Approach” by Invenção et al., for publication in the Vaccines, as part of the Special Issue entitled "New Approaches to Vaccine Development and Delivery".

All modifications are highlighted in the manuscript, and the response to reviewers follows below. All the authors confirm that they saw and agreed to the submitted paper. The authors have been recognized as contributors and have agreed to their inclusion. The material is original, and it has been neither published elsewhere nor submitted for publication simultaneously. None of the authors has any potential financial conflict of interest related to this manuscript.

 We appreciate the suggestions. The highlighted text has been revised.

Managing Editor

We hope this email finds you well. We need you to make some changes in the manuscript during revision. We noticed that the similarity rate is currently at 30% with other published papers. We need you to reduce the similarity by rephrasing the main text.

Response: We checked the manuscript, and the similarity presented corresponds to methodological terms mostly arranged in the sections corresponding to bioinformatics analyses, which are common in several scientific articles in the area. Therefore, similar excerpts that were reported do not constitute plagiarism of other published papers.

Reviewer 1

Accept in present form

Response: We appreciate the reviewer's suggestions and are pleased to address all concerns raised. I reiterate that the proposed comments have greatly improved the quality of our work, and we are very pleased with the discussions that have emerged.

Reviewer 2 Report

Comments and Suggestions for Authors

Regarding the three most serious defects of this study that I mentioned last time, the author did not make adequate revisions or supplements.

1.    After the DNA vaccine immunizes the body, it must first be demonstrated that the vaccine has been effective. The mechanism by which the DNA vaccine induces an immune response is not very clear, and there may be multiple pathways to activate immunity. The author suggests that the PCR method can be used to verify its expression. Does this imply that it can only induce an immune response in the form of nucleic acid? Those familiar with basic biology understand that PCR results can only confirm the presence of nucleic acid in the sample (or that the target gene has been introduced); they cannot prove that the target gene has been successfully expressed as a protein in the body. The target protein is also one of the initiators of the related immune responses. Subsequent studies are based on the assumption that the target gene has been introduced, expressions and functions (which may involve motifs, proteins, or other factors). However, the author did not adequately substantiate this foundational assumption.

2.    Almost all experiments should use biological experiments to validate bioinformatics results based on predictions. The sections from 3.1 to 3.4 of this study are entirely based on bioinformatics predictions, so I suggest conducting real biological experiments to verify the reliability of these predictions. However, the author evidently misunderstood my intention and only cited a number of literature sources to demonstrate that the prediction tool is reliable.

3.    This study also has significant flaws in its experimental design. As mentioned in my previous review, the study was designed to confirm the combined effect of S and N. The control group should include S alone and N alone, and also ideally a mixture of S alone and N (Non-fusion protein). While the results of this study can be discussed alongside findings from other studies, they still do not compensate for the flaws in the experimental design.

Reviewer 3 Report

Comments and Suggestions for Authors

The paper can be accepted